

# What doesn't kill them makes them stronger: an association between elongation factor 1-α overdominance in the sea star *Pisaster ochraceus* and "sea star wasting disease"

John P. Wares[1] and Lauren M. Schiebelhut[2]

[1] Department of Genetics and the Odum School of Ecology, University of Georgia, Athens, GA, United States

[2] School of Natural Sciences, University of California, Merced, CA, United States

## ABSTRACT

In recent years, a massive mortality event has killed millions of sea stars, of many different species, along the Pacific coast of North America. This disease event, known as 'sea star wasting disease' (SSWD), is linked to viral infection. In one affected sea star (*Pisaster ochraceus*), previous work had identified that the elongation factor 1-α locus (EF1A) harbored an intronic insertion allele that is lethal when homozygous yet appears to be maintained at moderate frequency in populations through increased fitness for heterozygotes. The environmental conditions supporting this increased fitness are unknown, but overdominance is often associated with disease. Here, we evaluate populations of *P. ochraceus* to identify the relationship between SSWD and EF1A genotype. Our data suggest that there may be significantly decreased occurrence of SSWD in individuals that are heterozygous at this locus. These results suggest further studies are warranted to understand the functional relationship between diversity at EF1A and survival in *P. ochraceus*.

## INTRODUCTION

One of the more stunning recent news stories pertaining to ocean health was the massive die-off of sea stars on both coasts of North America via a necrotic syndrome now known as sea star wasting disease (SSWD) (*Hewson et al., 2014*). Similar die-offs have happened in earlier decades (*Becker, 2006*; *Eckert, Engle & Kushner, 1999*), though none as extensive as in 2013–2014. *Hewson et al. (2014)* identified a candidate densovirus that is in greater abundance in diseased sea stars, and may be a causal agent; however, there is much yet to be learned. As sea stars are key predators in marine benthic ecosystems, the impacts of disease on these organisms could dramatically restructure coastal communities (*Paine, 1966*). Thus, we address here the potential for one species to harbor heritable variation and how this variation is affected by SSWD.

Corresponding author
John P. Wares, jpwares@uga.edu

During disease outbreaks, biologists are keen to know whether populations will exhibit any resistance to a pathogen. Thus, management studies may include surveys of genetic diversity to identify the potential for evolving resistance, or genetic rescue from other regions (*Whiteley et al., 2015*); such studies may also provide insight into the extent of population structure and gene flow among regions. Following a routine analysis of genetic variation in the sea star *Pisaster ochraceus* (*Harley et al., 2006*); *Pankey & Wares, (2009)* identified an insertion mutation in an intron of the elongation factor 1-$\alpha$ gene (hereafter EF1A) that appeared to exhibit overdominance. In this case, the insertion is lethal when homozygous (*Pankey & Wares, 2009*), yet the average frequency of the insertion allele was ~0.24 along the Pacific coast of North America. These observations suggest that the heterozygote has a significant fitness advantage in an unknown environmental setting. Overdominance is often associated with resistance to disease or toxins, however, and *Pankey & Wares (2009)*, referring to what is now called SSWD, speculated that:

"*widespread die-offs on the west coast of North America… could exert a substantial selective force on Pisaster. Given the prevalence of pathogen resistance in earlier studies of overdominance, we believe this to be a probable explanation for the maintenance of the described… polymorphism.*"

There is concern that elevated sea temperature is a component of the SSWD outbreak (*Bates, Hilton & Harley, 2009*; *Eisenlord et al., 2016*; *Hewson et al., 2014*). The relationship between expression of EF1A and thermal tolerance has been identified in other metazoans (*Buckley, Gracey & Somero, 2006*; *Stearns & Kaiser, 1993*), and functions in part through rapid co-production of proteins associated with the heat shock response. Though *Pankey & Wares (2009)* were not able to detect EF1A expression differences among individuals of differing genotype, we were not able at the time to control for a number of environmental factors nor the possible action of splice variants. Here, we posit an indirect mechanistic relationship between temperature, the effect of expression of EF1A, and SSWD. With as little as is known about this disease and marine disease in general (*Mydlarz, Jones & Harvell, 2006*), this is at best an educated guess. However, it is useful to know what potential *P. ochraceus* and other sea stars have for surviving this outbreak and natural patterns of genetic variation, and whether subsequent generations will be more resistant or tolerant of similar pathogens. Here we evaluate this simple polymorphism from populations of *P. ochraceus* collected prior to and following the SSWD outbreak, as well as focus on extant individuals and their disease status. We ask whether there are frequency shifts of the two genotypes at this locus that may be associated with resistance to SSWD, and evaluate efforts to explore similar genomic variation in other species.

## METHODS

### *Pisaster* and disease status

Collections were made in 2014–2015 from locations in central California, Oregon, coastal Washington and the San Juan Islands (WA), and Nanaimo (Vancouver), and categorized by health status using the Pacific Rocky Intertidal Monitoring Network

**Table 1** Sample sizes from each regional collection of individuals (see Table S1 for additional sampling information); samples are listed by health status as well as genotype (+/+ *wild type*, +/ins for the heterozygote genotype). Effect size refers to the difference in proportion of EF1A homozygotes that exhibit SSWD and the proportion of heterozygotes with SSWD; positive numbers suggest a higher proportion of homozygotes with SSWD.

| Site/region | Latitude | Longitude | SSWD symptomatic +/+ | SSWD symptomatic +/ins | SSWD asymptomatic +/+ | SSWD asymptomatic +/ins | Effect size |
|---|---|---|---|---|---|---|---|
| Nanaimo, Vancouver, BC | 49.2 | 124 | 8 | 4 | 7 | 5 | 0.089 |
| Olympic Peninsula, WA | 48.5 | 125 | 5 | 0 | 11 | 4 | 0.31 |
| San Juan Island, WA | 48.5 | 123 | 25 | 17 | 15 | 18 | 0.14 |
| Cape Meares, OR | 45.5 | 124 | 1 | 0 | 4 | 5 | 0.2 |
| Seal Rock, OR | 44.5 | 124.1 | 1 | 2 | 1 | 6 | 0.25 |
| Coquille Point, OR | 43.1 | 124.4 | 7 | 0 | 1 | 2 | 0.88 |
| Damnation Creek, CA | 41.7 | 124.1 | 1 | 1 | 4 | 4 | 0 |
| Sonoma County, CA | 38.7 | 123.4 | 8 | 1 | 22 | 18 | 0.21 |
| San Francisco Bay, CA | 38.0 | 122.8 | 0 | 0 | 7 | 13 | 0 |
| **Overall** | | | **56** | **25** | **72** | **75** | **0.19** |

classification (http://www.eeb.ucsc.edu/pacificrockyintertidal/index.html; Table 1). Complete information on collection location, individual sizes, permitting, and other metadata are in Data S1. All permits are listed in Supplemental Information 1. Specimens were collected under California Department of Fish and Wildlife permits #11794 to L Schiebelhut and #603 to MN Dawson, California State Parks permit to MN Dawson, and National Parks permits PORE-2012-SCI-0038, PORE-2013-SCI-0033, and REDW-2014-SCI-0018. Collections at Nanaimo were under Dept. of Fisheries and Oceans site permit at Pacific Biological Station; tissues from the San Juan Islands were re-sampled under permits for *Eisenlord et al. (2016)*.

## Temporal comparisons

In addition to individuals explicitly assessed for health status, we also considered the potential for genotype (and allele) frequency change following a related disease outbreak. Previous EF1A genotype/allele frequency information from specimens collected in 2003–2004 are available for many locations along the Pacific coast (*Pankey & Wares, 2009*). In central California, tissues from four locations (Sonoma County) were obtained in both 2012 (pre-outbreak) as well as 2014 (the tissues noted in previous section from these sites). Thus, genetic frequencies from three time points can be assessed. Along the Oregon, Washington, and San Juan Island coastal regions, the tissues noted in previous section (from 2014 to 2015) can be compared to the genetic frequency information from 2003 to 2004 tissues.

## Molecular methods

As in *Pankey & Wares (2009)*, primers PisEF1-F (5′-*aggctgccgataccttcaa*-3′) and PisEF1-R (5′-*gctagtatctgtttctgtgtgactgc*-3′) were used to determine individual EF1A genotypes by scoring length-polymorphic PCR products on 2% (or greater) agarose gels. About 10% of individuals were multiply genotyped so that genotype error rate (*Pompanon et al., 2005*) could be assessed. An example of this polymorphism is shown in Fig. 1.

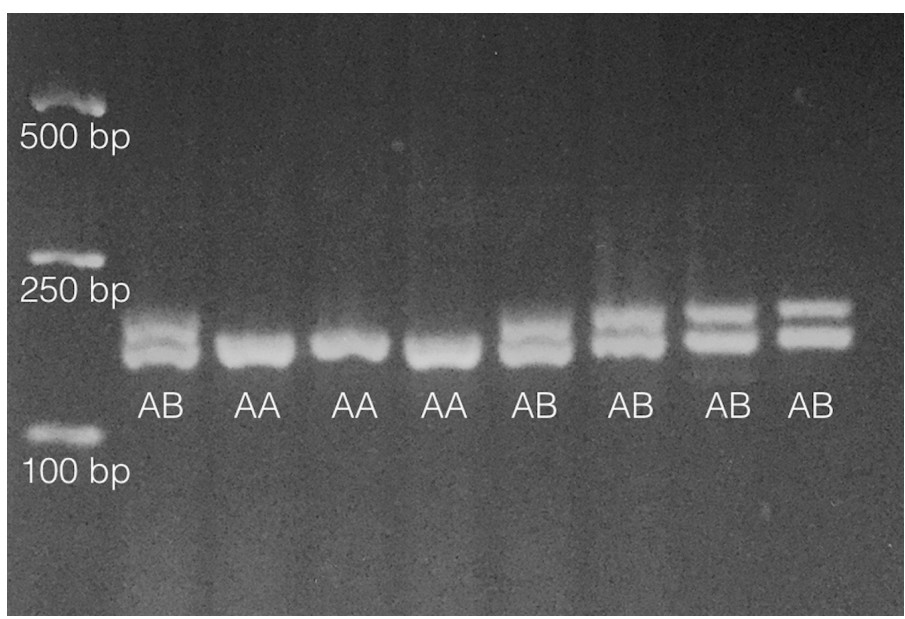

**Figure 1 Results from analysis of 8 individuals on 2% agarose gel following PCR amplification as noted in 'Methods.'** Fragments only vary by 6bp in length so gel must be run for ∼60 min under typical conditions (∼90–100 V). Heterozygotes are denoted 'AB' and homozygotes denoted 'AA' on gel image. Size standards are on left side of gel.

## Statistical analyses

Our first approach is to ask whether the frequency distribution of the two EF1A genotypes differs between diseased and healthy individuals of *P. ochraceus*. Specimens from distinct locations are grouped when sites are within 50 km of each other and each regional sample is evaluated separately as well as combined. Separate analyses recognize the potential for heterogeneity at related quantitative traits despite apparent phylogeographic homogeneity (*Harley et al., 2006*), as well as distinct environmental influences, while pooled analysis augments statistical power.

Regional and combined data are analyzed first with a Fisher's exact test. Additionally, following *Gerrodette (2011)*, we estimate the effect size of genotype on SSWD mortality by subtracting the proportion of homozygous individuals exhibiting SSWD from the proportion of heterozygous individuals exhibiting SSWD. Here we assume that the genotype frequencies are binomially distributed (with associated sampling error) and estimate the difference in proportion of disease incidence (also here assumed binomial) between homozygotes and heterozygotes. Again, these probability distributions are estimated for each individual regional/temporal sample. These same statistical methods were applied to evaluate genotype frequency changes between population samples from before and after the 2013–14 SSWD outbreak, as described above.

To evaluate the probability that heterozygous individuals have a higher probability of avoiding or surviving SSWD, logistic regression of the complete dataset is performed with models that incorporate individual size (when available, measured from center of disk to

tip of an arm), genotype at EF1A, and region of collection. Models of single factors were evaluated, and for factors that exhibit significant variation additive and interaction models were also evaluated and compared using AIC values. The model with greatest AIC weight is considered to explain most of the variation in the system. All statistical analyses were performed using R (*R Development Core Team, 2009*).

## RESULTS

The genotype error rate was 0 for 21 individuals (out of 228 in this study) that were repeatedly genotyped. Two individuals initially presented a very faint second band on gel, but subsequent repeat amplification of one of these confirmed it as homozygous (note: this polymorphism has been assessed as an overdominant Mendelian locus (*Pankey & Wares, 2009*), so we do not think this represents variation in paralog amplification success).

Data from each location (Table 1) were analyzed with a basic Fisher's exact test. Each regional sample was in the predicted direction with a higher likelihood of SSWD among homozygotes than heterozygotes; no single location or regional grouping presented contrasts that were statistically significant (results not shown). Certainly a component of individual regional/temporal samples is that with modest sample sizes, statistical power is low. Combining all samples leads to a statistically significant result ($p$-value 0.0035).

When considering all available information related to disease risk in our samples, models representing single factors (size, sample location, and genotype) and combinations of factors were compared using AIC against a null (no factor) model. Size (radius) was not significant and was dropped from subsequent analyses, while location ($p < 0.001$) and genotype were both significant factors ($p = 0.00615$). Given the remaining main effects and interactions, we found the AIC weight was strongest for models with both factors included (AIC weight 0.226) and with both factors included with interaction between (AIC weight 0.756). When both factors are included, both are still significant ($p < 0.01$); when the model includes interactions, genotype is important but the additional parameterization reduces power ($p = 0.0526$), location is not significant, and the interaction term is not significant ($p = 0.0694$). A small sample of individuals ($n = 14$) had no size measurement available (Supplemental Information 1) but exhibited no significant effect (effect $= 0.05$, $p = 1$) of genotype.

Despite the hypothesis of increased fitness for EF1A heterozygotes under these conditions, the frequency of the insertion (*ins*) allele in central California only appears to decrease through time, from approximately 0.27 ($n = 33$) in 2003–2004 (*Pankey & Wares, 2009*) to 0.24 ($n = 40$) in 2012 and 0.25 ($n = 40$) in 2014–2015. However, with sampling error these frequencies are statistically unchanged and a larger sample comparison may be necessary to explore this component of our evaluation. In the San Juan Islands, the frequency of the *ins* allele is also statistically unchanged from 0.27 in 2003 ($n = 62$) to 0.253 in 2014 ($n = 75$), again not supporting a hypothesis of selection increasing or maintaining the *ins* frequency (but also not a statistically significant difference in frequency). The same can be said for contrasts along the Oregon and Washington coasts, where the overall frequency is effectively unchanged.

## DISCUSSION

Our data show that our hypothesis for a relationship between disease status (SSWD) and an apparent overdominant polymorphism in *P. ochraceus* is strongly supported—results from each sample are in the predicted direction, and overall there is clear evidence that sick individuals are more likely to be EF1A homozygotes than heterozygotes. The net effect size of 0.19 (Table 1) is very similar to the fitness differential between genotypes (~0.2) proposed by *Pankey & Wares (2009)* given simulations of overdominance. The variation in effect seen among regional samples—and the apparent statistical interaction between sample location and genotypic effect on disease status—is likely influenced both by modest sample sizes as well as distinct exposure histories. It is likely that each of our regional samples has been exposed to distinct temperature profile histories (*Bates, Hilton & Harley, 2009*; *Eisenlord et al., 2016*), and it is possible that despite apparent population genetic homogeneity (*Harley et al., 2006*) that undetected evolutionary changes have led to distinct reaction norms among regional samples. Additionally, the timing of arrival and effect of the candidate densovirus that may cause SSWD may lead to distinct evolutionary dynamics across regions.

In previous work (*Bates, Hilton & Harley, 2009*; *Hewson et al., 2014*) it has been evaluated whether size was a predictive factor in disease status; both research groups found no clear statistical association (though it appeared there is a negative relationship between densovirus abundance and size in *Pycnopodia helianthoides*; *Hewson et al., 2014*). A more recent analysis (*Eisenlord et al., 2016*) does show a significant relationship between elevated temperature exposure, as well as size, and SSWD in *P. ochraceus*. Our smaller sample presents no clear association between sea star radius and disease status nor an interaction with EF1A genotype.

### Evolutionary response

Despite the apparent and predicted effect in our samples, we do not see the hypothesized evolutionary response—the frequency of the *ins* mutation has apparently not increased in recent years. If EF1A in *P. ochraceus* truly evolves via overdominance, where the heterozygote is significantly more fit under certain environmental conditions, then we would expect this allele to increase in frequency when exposed to a relevant mass mortality event. However, it is also not entirely clear what proportion of individuals have died in recent years as a result of SSWD (though estimates from intertidal surveys are high enough that some frequency response is warranted; see *Eisenlord et al., 2016*). Thus, detection of this change could be masked in part by simple stochastic changes (e.g., genetic drift) in local populations of *P. ochraceus*, and the sample size available from all three time points limits our statistical power to address this hypothesis. Of course, there are other forms of mortality in sea stars like *P. ochraceus* (*Jurgens et al., 2015*), and so it is still likely that we are seeing an indirect interaction between recent die-offs and individual-level responses that appear to be genotype dependent. Preliminary evidence (L Schiebelhut, 2015, unpublished data) suggest higher frequency of the *ins* allele at some California sites, and some of our sampled locations had higher frequencies of SSWD after tissues were harvested. Thus further study,

tracking frequency of the EF1A *ins* polymorphism through time with detailed information on disease and other environmental factors at local sites, is warranted.

## Disease in sea stars

With limited understanding of immune response in most echinoderms (*Mydlarz, Jones & Harvell, 2006*), the problem of SSWD is difficult enough to explore in *P. ochraceus*, let alone the many other species affected in the recent outbreak. In another species of sea star (*P. helianthoides*), *Fuess et al. (2015)* have identified some of the genomic components that are upregulated in response to viral exposure; however, we know of no similar (apparent) overdominant system in these other asteroids, and as yet still know very little about how this polymorphism in *P. ochraceus* influences EF1A expression, alternate splicing events, or what genes may be linked to this region and thus affected. Although some studies have indicated a relationship between elongation factor expression and thermal tolerance in plants (e.g., *Bukovnic et al., 2009*) there is still little information in metazoans (*Buckley, Gracey & Somero, 2006*; *Stearns & Kaiser, 1993*; *Stearns, Kaiser & Hillesheim, 1993*) though these studies suggest that differential expression is associated with increased lifespan and increased tolerance to heat stress. As this gene is considered "housekeeping" in many studies it is actually used to normalize the differential expression results of other genes (*Fangue, Hofmeister & Schulte, 2006*); our study suggests further attention to differential expression of this locus is warranted.

Our interest in exploring this particular case has little to do with solving the problem of disease, and more about the question of what demographics will be like for *P. ochraceus* in an increasingly warmer—and disease-affected—environment (*Eisenlord et al., 2016*; *Harvell et al., 2002*). If disease like SSWD interacts with the EF1A polymorphism as noted here, and the frequency of the deleterious *ins* allele increases, this could also indicate increased reproductive loss through homozygous lethality, which could decrease the potential for populations to rebound from crashes.

## Parallels with malaria

*Aidoo et al. (2002)* note that "sickle cell trait" in humans (carrying a single *S* allele of hemoglobin) provides ∼60% protection against overall mortality, mostly in the first 16 months of life; being a carrier is not a guarantee against infection. Other studies have focused on specific malarial parasites, and note that children heterozygous for the *S* hemoglobin allele have approximately one-tenth the mortality risk from *Plasmodium falciparum* as those homozygous for normal alleles (*Cholera et al., 2008*). In the absence of cohort data, it is difficult to estimate the level of disease or mortality protection that any single polymorphism can provide (*Aidoo et al., 2002*). Until such recent studies, the claim of overdominant selection on hemoglobin genotypes, based on the relationship between the frequency of the *S* allele and the prevalence of malaria (*Allison, 1954*), was only correlative. This relationship is still a clear case of overdominance (*Gemmell & Slate, 2006*), but is illustrative that increased heterozygote fitness does not require absolute protection against the associated risk factor—e.g., that all individuals of *P. ochraceus* with SSWD would be homozygous for the wild-type EF1A allele identified in

*Pankey & Wares (2009)*, nor that healthy individuals would all be heterozygotes—as there are many components to disease avoidance, tolerance, or resistance.

## CONCLUSIONS

At this time, with limited sampling (and recognizing that our samples themselves may not be random from populations of *P. ochraceus*), our results suggest an intriguing (but probably indirect) relationship between SSWD susceptibility and the EF1A polymorphism described. The direction of effect is consistent in *all* subsamples, and the magnitude of effect overall is comparable to predictions based on simulations of overdominance in this system given the observed frequency of the *ins* allele and lethality of *ins* homozygotes (*Pankey & Wares, 2009*). Nevertheless, we do not see an increase in the frequency of the *ins* allele over time in our samples and so we remain curious about the dynamics of overdominance in this system.

It is possible that regulation and expression of EF1A is influenced by this polymorphism in a way that alters an individuals' tolerance or capacity for heat stress, and in a warming climate and ocean it is known that disease and mortality are higher in large part because of physiological stress modifying an organism's response to pathogens (*Eisenlord et al., 2016*; *Harvell et al., 2002*). Further work is needed not only to examine the association shown here, but also to identify (i) whether size or maturity is truly important in this relationship, (ii) whether individuals of different genotype do have distinct constitutive or regulated patterns of expression of EF1A or related/linked genes, and (iii) whether there are genotype-driven differences in mortality of individuals under thermal stress (which can affect feeding rates as well as physiological factors in *P. ochraceus*; *Sanford, (2002)*).

In the meantime, we emphasize that this system is so easy to explore as a low-budget research project or teaching tool that there are opportunities to work as a community to greatly expand our understanding of the maintenance of the overdominant EF1A diversity in *P. ochraceus*, perhaps for other pertinent variables of interest. We would encourage any future studies to ensure that sufficient metadata are associated with any such comparison so that this relationship can continue to be explored.

## ACKNOWLEDGEMENTS

The authors would like to thank Collin Closek (Wares Lab alumnus), Mo Turner, and Morgan Eisenlord for coordinating tissue samples from FHL; Mike Hart, Vanessa Guerra, and David Breault for specimen collection and handling of Nanaimo tissues in 2015; and Peter Raimondi and PISCO for other coastal samples collected in 2014–2015. We thank Sarah Abboud, Charlsie Berg, Anny Calderon, Lorely Chavez, Brendan Cornwell, Michael N. Dawson, Madlen Friedrich, Brian Gaylord, Alehandra Guzman, Brittany Jellison, Laura Jurgens, Shawn Knapp, Kelly McClintock, Holly Mondo, Mira Parekh, Emily Ramirez, Mariana Rocha de Souza, Adam Rosso, Stephen Sanchez, Holly Swift, Sabah Ul Hussan, and Jesse Wilson for assisting with California collections in 2012–2014. Tim Makinde generated much of the genotypic data at UGA. Members of the Wares Lab at UGA and colleagues at the Odum School of Ecology, particularly John Drake, assisted greatly with discussion of the

idea and early drafts of the manuscript. The manuscript also benefited from constructive comments by Morgan Eisenlord, Associate Editor Keith Crandall, and three anonymous reviewers. We also thank Sarah Gravem for organizing the SSWD-themed session at the Western Society of Naturalists meeting in 2015 from which we gained perspective and useful comments on this project.

### Funding

Funds for this work come from California Sea Grant College Program grant #2012-R/ENV-223PD and National Science Foundation grant OCE-1243958 and OCE-1243970 (LMS), as well as National Science Foundation OCE-1015342 (JPW). The funders had no role in study design, data collection and analysis, decision to publish, or preparation of the manuscript.

### Grant Disclosures

The following grant information was disclosed by the authors:
California Sea Grant College Program: #2012-R/ENV-223PD.
National Science Foundation: OCE-1243958, OCE-1243970, OCE-1015342.

### Competing Interests

The authors declare there are no competing interests.

### Author Contributions

- John P. Wares conceived and designed the experiments, performed the experiments, analyzed the data, contributed reagents/materials/analysis tools, wrote the paper, prepared figures and/or tables, reviewed drafts of the paper.
- Lauren M. Schiebelhut performed the experiments, analyzed the data, contributed reagents/materials/analysis tools, wrote the paper, prepared figures and/or tables, reviewed drafts of the paper.

### Field Study Permissions

The following information was supplied relating to field study approvals (i.e., approving body and any reference numbers):

All permits are listed in Supplemental Information 1. Specimens were collected under California Department of Fish and Wildlife permits #11794 to L Schiebelhut and #603 to MN Dawson, California State Parks permit to MN Dawson, and National Parks permits PORE-2012-SCI-0038, PORE-2013-SCI-0033, and REDW-2014-SCI-0018. Collections at Nanaimo were under Dept. of Fisheries and Oceans site permit at Pacific Biological Station; tissues from the San Juan Islands were re-sampled under permits for *Eisenlord et al. (2016)*.

### Data Availability

The complete data set for this manuscript is provided as Supplemental Information.

## Supplemental Information

Supplemental information for this article can be found online at http://dx.doi.org/10.7717/peerj.1876#supplemental-information.

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
