# Peer review of "What doesn’t kill them makes them stronger: an association between elongation factor 1-α overdominance in the sea star Pisaster ochraceus and “sea star wasting disease”"

_PeerJ, doi:10.7717/peerj.1876_

## Round 0.1 · original submission · Minor Revisions

Your paper has now been reviewed by three reviewers, all of whom found your paper very interesting, well written, and appropriate for the journal. They have all noted a number of minor concerns. Therefore, I am recommending minor revisions for you to consider. Please pay detailed attention to their extensive list of issues.

Reviewer 1 ·

Basic reporting

Overall the manuscript conforms to the journal standards, with on exception. Table 1 appears to contain results, and thus should be moved to that section. The final predicate, relating to crowd-sourcing of future work, may also not be in line with journal's policy.

Experimental design

The question of susceptibility to SSWD based on genetic difference is a massive issue in the current wasting disease, especially in Pisaster ochraceus, where some populations appear heavily affected, yet others within close proximity appear less affected. The relationship between SSWD and the candidate densovirus SSaDV is also weakest for this species, suggesting either that SSaDV is more weakly associated with the disease, or that there may be some kind of resistance in the population. I found this contribution to be highly relevant to the overall story of SSWD and its potential causes. The experimental design appears to be sufficient to address the hypotheses posed by the authors. All permits are appropriate to the sample collection.

Validity of the findings

This manuscript presents an interesting observation of EF1A genetic variation and its relationship to wasting disease occurrence in North American west coast sea stars. The authors had previously observed EF1A variation in sea stars, identifying a lethal insertion when homozygous which may be linked to enhanced fitness when heterozygous. The authors applied this knowledge to a suite of samples collected from California, Oregon and Washington, as well as alleles from previous studies (2003 – 2004, 2012 and present) to compare frequencies over time (before/after the current disease outbreak), between SSWD and asymptomatic stars, and between locations, taking into account location and individual size. Overall the authors found that SSWD occurrence was associated with heterozygosity in the EF1A locus, but there was no significant increase in the EF1A ins frequency before and after the disease event.

The findings appear to be valid, in that they link EF1A heterozygosity with SSWD occurrence and thus, potentially, fitness. I do however, feel that there are places in the manuscript regarding thermal tolerance which need to either be reduced (especially in the discussion) or eliminated altogether. The manuscript presents no information about temperature relationship, yet this is the single factor not directly tied to the study which is presented (i.e. page 5, para 3). I think that this should be toned down as there are probably many other environmental parameters (pH, algal blooms, salinity) that may equally participate in SSWD susceptibility.

Additional comments

Below I indicate specific comments about the manuscript which could improve on the reporting:
Abst, line 1: "plague" seems an odd choice of words. I'd suggest "mortality event" may be more appropriate.
Abst, line 2: "disease" may be better as "event"
Abst, line 3: "is though to be caused by" is better "is linked to" - there is, as yet, almost no evidence that the virus is the cause of the disease - only an association.
Abst, line 3: "in the affected stars" might be better as "in one of the affected star species" - there are > 20 species affected, of which P. ochraceus is one.
Abst, line 9: "P. ochraceus to identify" might be better as "P. ochraceus populations affected by SSWD to identify"
Abst, line 10: "decreased infection and mortality rates" - as there is as both infection and mortality rates are not determined in this study, this may be better worded as "occurrence of SSWD"
Abst, last line: there appears to be something missing from the last line here, perhaps "are needed to elucidate the mechanism by which EF1A and SSWD are related" or something like that.
Page 4, Line 41: "one species to respond to disease" suggests a dynamic interaction. "be affected by" or "be related to" may be more appropriate.
Page 5, Line 59-60: It is unclear what the "this is a best educated guess" - if this relates to the previous paragraph, then a pgh break shouldn't be there.
Page 5, Line 66: "resistence to infection" should be "resistence to SSWD" as infection and SSWD are not synonymous (it's unclear whether it is infectious as yet.
Table 1 should be in the results section, as it contains results. "SSWD" and "OK" should be replaced by "SSWD symptomatic" and "asymptomatic" respectively. The authors should include columns for SSWD and OK for the total number of individuals tested.
Page 7, Line 107: "under typical conditions" - those electrophoretic conditions should be spelled out so that other scientists can follow.
Page 9, Line 180: "temperature profile histories" - amongst many other possible environmental parameters. I suggest that the authors drop this statement as there is no investigation in this study between temperature and EF1A allele frequency.
Page 9, Line 183: "densovirus that causes SSWD" - there is no convincing data to suggest definitively that SSaDV causes SSWD (yet). I suggest " candidate densovirus that may cause SSWD".
Page 9, Line 190: Recent suggestions from the press are that SSWD is less frequently found in juveniles than in adults, though there may be a threshold affect whereby above a certain size the younger adult stars may be more susceptible. The jury, as best I can tell is out.
Page 10, Line 223: Thermal relations with disease are reported numerous times in this manuscript. In my mind, since the authors do not investigate any relationships with temperature, these statements, like here, should be reduced or removed.
Page 11, Line 255: Here especially, when referring to thermal tolerance and EF1A, it is entirely speculative. The start of this paragraph should be removed. It detracts from the interesting observation of EF1A heterozygosity and SSWD occurrence!
Page 11, Line 270: This last part of the sentence is disposable. I have never read a statement attracting crowdsourcing to perform analyses in any journal. I strongly recommend this dropped (in fact it may not conform to journal's policy).

Reviewer 2 ·

Basic reporting

The authors met these criteria for basic reporting.

Experimental design

Generally appropriate, although see comments below.

Validity of the findings

This manuscript presents an intriguing link between an insertion polymorphism in a housekeeping gene (elongation factor 1a) and the symptoms of a viral disease that has recently caused massive dies of an ecologically-important sea star. The meat of the paper lies in Table 1, which presents the frequencies of heterozygotes with the insertion and homozygotes without the insertion (all homozygotes receiving two insertion alleles apparently die). Most of the populations are poorly sampled, as witnessed by the genotypic shifts were not significant by simple contingency table analyses (but see note on San Juan Island below). Despite this weakness, I do agree with the authors that the direction of the shift is in the expected direction for most populations (i.e., heterozygotes are more common in healthy than in sick individuals), and the overall logistic regression model does support the contention.

The temporal comparison is also admittedly weak, but the authors are appropriately cautious about its interpretation.

Additional comments

i thought of an alternative hypothesis that the authors may want to consider: EF1 polymorphism may reflect greater heterozygosity across the genome, given there is a great deal of literature on links between genome-wide heterozygosity and fitness. do you have any data from other nuclear loci to address this?

line 23: running chain of modifiers…maybe remove?

line 60: what does “this” refer to exactly?

is san juan island significant (line 191) or not (line 43)?

line 160: do not say this allele frequency decreased anywhere in this paragraph, since you have no statistical support. it remained unchanged.

this raises an issue about what should you expect from ins allele frequency. let’s say that everyone that has the disease dies (although i understand that they don’t necessarily do so). you show here that populations move from p = 0.222 (before the disease (i calculated this using the column sums in table 1) to p = 0.255 in a single generation. this says to me that you definitely don’t have the power to detect this minor a shift at all, which means i would be even more circumspect than you are in this draft.

Reviewer 3 ·

Basic reporting

This is an interesting and timely report that makes an intriguing case for overdominance as a potential mechanism offsetting one of the defining non-human disease outbreaks of our time, the mass die-off of Pacific sea stars. In the context of global change, such outbreaks are projected to increase in severity and frequency, and for ecologically pivotal species like Pisaster, there are ecosystem-wide implications for patterns of disease resistance and evolvability of resistance. Against that background, this paper makes an important contribution and builds appropriately on earlier work in the system, and is suitable for publication in PeerJ.

Experimental design

Appropriate; it would be nice to see a table presenting the different models tested and their AIC scores, however. Also, Methods could better explain the calculation of "effect" or benefit conferred by genotype that is presented in Table 1, which is explained only in the legend and not in much detail. The authors could also comment on why their sample sizes were low - surely there's plenty of pre-outbreak tissue in freezers and museums to draw upon, but perhaps not? Also, a bit of background on San Fran Bay -- did the disease not hit hard there, or did it hit so hard there's no surviving sick Pisaster to sample?

Validity of the findings

Clear and compelling; I appreciate the explanation of comparative fit of different models.

Additional comments

Overall, I would appreciate a little background on the EF2a gene itself: What does this gene do, and why might it play a role in disease? The intro set me up to expect some discussion of this later in the paper, including why temperature and alternative splicing might be involved here, but there was no return to this topic or payoff on that front. I would have appreciated a little more attention to any possible mechanistic basis for overdominance at the end of the paper, and some consideration of why this gene might be important.

I haven't thought about this in much depth, but I wonder about the lack of observed increase in frequency for the insertion over time. It seems like it was initially at a VERY high frequency during a non-outbreak period, for a homozygous lethal allele. Could it be that this allele is maintained by balancing selection at ~0.25 and that this is the highest frequency is can reasonably reach, before the fitness costs outweigh any benefit? I wonder if heterozygotes gain a generic fitness advantage, which in turn makes them more likely to survive wasting disease, rather than specifically gaining disease resistance per se (a-la sickle cell trait) -- these seem like alternative hypotheses that warrant some consideration here. The surprise to me is less the lack of an increase in frequency so much as the starting high frequency for something that otherwise should be STRONGLY selected against by background conditions.

The "conclusions" section is really more like "future directions" than a summary of the main findings, and might be appropriately retitled as such. "Organisms" in that section is missing an apostrophe (possessive not plural).

---

## Round 0.2 · accepted · Accept

Thanks for your careful attention to the many comments by all three reviewers and your explicit response to each comment. I think your revisions are well done and am happy to recommend acceptance at this point.